# Advancing *Bacillus licheniformis* as a Superior Expression Platform through Promoter Engineering

**DOI:** 10.3390/microorganisms12081693

**Published:** 2024-08-16

**Authors:** Fengxu Xiao, Yupeng Zhang, Lihuan Zhang, Siyu Li, Wei Chen, Guiyang Shi, Youran Li

**Affiliations:** 1Key Laboratory of Industrial Biotechnology, Ministry of Education, School of Biotechnology, Jiangnan University, Wuxi 214122, China; 8202306022@jiangnan.edu.cn (F.X.); 8202407007@jiangnan.edu.cn (Y.Z.); 13115410926@163.com (L.Z.); 7210201012@stu.jiangnan.edu.cn (S.L.); gyshi@jiangnan.edu.cn (G.S.); 2National Engineering Research Center for Cereal Fermentation and Food Biomanufacturing, Jiangnan University, 1800 Lihu Avenue, Wuxi 214122, China; 3Jiangsu Provincial Engineering Research Center for Bioactive Product Processing, Jiangnan University, Wuxi 214122, China; 4School of Food Science and Technology, Jiangnan University, Wuxi 214122, China; weichen@jiangnan.edu.cn

**Keywords:** *Bacillus licheniformis*, promoter, promoter engineering

## Abstract

*Bacillus licheniformis* is recognised as an exceptional expression platform in biomanufacturing due to its ability to produce high-value products. Consequently, metabolic engineering of *B. licheniformis* is increasingly pursued to enhance its utility as a biomanufacturing vehicle. Effective *B. licheniformis* cell factories require promoters that enable regulated expression of target genes. This review discusses recent advancements in the characterisation, synthesis, and engineering of *B. licheniformis* promoters. We highlight the application of constitutive promoters, quorum sensing promoters, and inducible promoters in protein and chemical synthesis. Additionally, we summarise efforts to expand the promoter toolbox through hybrid promoter engineering, transcription factor-based inducible promoter engineering, and ribosome binding site (RBS) engineering.

## 1. Introduction

Generally recognised as safe (GRAS), *B*. *licheniformis* is a Gram-positive bacterium that produces endospores, a trait within the *Bacillus* genus [1]. Natural habitats for *B. licheniformis* include (1) soil; (2) the ocean; (3) bird feathers; and (4) plants [2,3,4]. In natural settings, *B. licheniformis* can stimulate plant growth and enhance crop tolerance to diseases by activating the jasmonic acid/ethylene signalling pathway [5,6,7]. Additionally, *B. licheniformis* synthesises various secondary metabolites that function as insect hormones, such as ethylene glycol and 2,3-butanediol, which promote insect mating and enrich environmental carbon compounds [8]. Research on *B. licheniformis* began in 1945, but gained significant momentum only when its advantages in amylase production and biosafety were recognised [1].

Recently, *B. licheniformis* has become popular as a cell factory for producing high-value enzymes like amylase, arginase, and amylosucrase [9,10,11], as well as high-value chemicals, such as acetoin and 2,3-butanediol [12,13] (Figure 1). It can grow on low-cost carbon substrates like sucrose, maltose, and starch [14,15,16]. A recent study demonstrated that engineered *B. licheniformis* could utilise marine algae biomass rich in sulphated polysaccharides from green algae as a growth substrate [17]. This underscores *B. licheniformis’*s unique advantages in using diverse carbon substrates. Moreover, it can thrive at high temperatures [18,19]. During fermentation, *B. licheniformis* produces antimicrobial substances like bacteriocins, reducing susceptibility to bacterial infections [20]. These attributes have led to its widespread use in modern fermentation industries.

The rapid advancement of synthetic biology technologies and genomic sequence analysis has accelerated metabolic engineering in *B. licheniformis*. In synthetic biology, promoters are crucial for regulating carbon flow allocation and target gene expression. Selecting appropriate promoters is the first step in expanding *B. licheniformis’*s applications. Natural promoters generally fall into three categories: constitutive promoters, inducible promoters, and quorum sensing (QS) promoters (Figure 2). Constitutive promoters maintain steady expression levels regardless of external stimuli; inducible promoters are activated by specific inducers; quorum sensing promoters adjust expression based on bacterial density. Xylose-inducible promoters have been utilised in *B. licheniformis’*s CRISPR gene editing methods [21]. Typically less than 200 bp long, *B. licheniformis’*s promoters consist of an upstream regulation region and a core promoter region. The core region contains crucial sites for RNA polymerase recognition and binding, while the upstream regulation region includes specific transcription factor recognition sites. The diversity of artificial reprogramming promoters has increased due to mechanistic analyses of key transcription factors (DegU, AbrB, CcpA, and GlnR) involved in *B. licheniformis’*s life activities [22,23,24,25]. Customisable artificial promoters can be created by identifying these transcription factor recognition sites and incorporating them into constitutive promoters. Typically, these synthetic promoters exhibit higher thresholds and novel inducibility.

In 2023, the strategy of using *B. licheniformis* as a cell factory to produce high-value chemicals was published [26]. However, there is a lack of comprehensive understanding regarding the characteristics and engineering strategies of *B. licheniformis* promoters, particularly concerning their applications in chemical biosynthesis or protein synthesis. This review aims to provide an overview of the advancements made in the study of *B. licheniformis* promoters. We demonstrate the application of constitutive promoters, quorum sensing promoters, and inducible promoters in protein or chemical synthesis. Additionally, we propose promoter engineering strategies to expand the promoter library of *B. licheniformis*, including (1) hybrid promoter engineering, (2) inducible promoter engineering based on transcription factors, and (3) ribosome binding site (RBS) engineering.

## 2. Constitutive Promoters

### 2.1. Endogenous Strong Constitutive Promoters

There are two methods for obtaining *B*. *licheniformis’*s endogenous strong constitutive promoters: (1) mining from strong metabolic pathways and (2) mining from transcriptome data. The two main strong metabolic processes are the 2,3-butanediol/acetoin synthesis pathway and the bacitracin synthesis pathway.

#### 2.1.1. P*_bacA_* Derived from Bacitracin Synthase Operon

*B*. *licheniformis* secretes bacteriocins, which inhibit the growth of Gram-negative bacteria [27]. The *bacT* and *bacABC* components of the *B. licheniformis* peptide synthase operon were initially described in 1997 [28]. The *bacT* gene encodes thioesterase, while the *bacABC* operon encodes non-ribosomal peptide synthase (NRPS). The promoter P*_bacA_*, responsible for the transcription of the *bacABC* operon, is a strong endogenous promoter in *B. licheniformis*. Overexpressing *ilvBHC*, the *leuABCD* operon, *ilvD*, the leucyl-tRNA synthase gene *leuS*, and the pulcherriminic acid synthase cluster *yvmC*-*cypX* through the promoter P*_bacA_*, and knocking out the gene *bkdAB*, achieved pulcherriminic acid yields of 507.4 mg/L, which were 337.8% higher than those of the starting strain [29]. Zhan et al. substituted the P*_bacA_* promoter for the P*_glpFK_* promoter in the *B*. *licheniformis* glycerol operon, leading to an 18.8% rise in glycerol consumption over the wild strain [30]. Shi et al. substituted the native promoter of the bkd operon with P*_bacA_*, which increased the production of short-chain fatty acids (SBCFAs) to 4.68 g/L, a 1.98-fold increase. The modified strain produced 8.37 g/L of SBCFAs in a 5 L fermentation tank, yielding 0.20 g/L/h [31].

#### 2.1.2. P*_alsSD_* Derived from alsSD Operon

The two primary overflow metabolites of *B*. *licheniformis* are 2,3-butanediol and acetoin [32]. The P*_al__s__SD_* promoter regulates the *alsSD* operon, a crucial operon that catalyses the transcriptional conversion of pyruvate to acetoin. Wu et al. selected the top 10 promoters with the largest upregulation fold for study based on transcriptome data. They used green fluorescence as the reporter gene to show that P*_alsSD_* had significantly higher activity than the other nine promoters [20].

### 2.2. Heterologous Strong Constitutive Promoters

#### 2.2.1. P_43_

The P_43_ promoter derived from *Bacillus subtilis* is widely used in *B. licheniformis*. Cai et al. assembled the P_43_ promoter, the *aprE* signal peptide, and nattokinase to create an expression cassette for nattokinase. Furthermore, 35.60 FU ml^−1^ of nattokinase activity was obtained by overexpressing the signal peptidase SipV [33]. In *ccpA*-deficient strains, Zhang et al. overexpressed the *ccpA* gene via the P_43_ promoter, enabling recombinant *B*. *licheniformis* to utilise glucose and xylose simultaneously [34]. Li et al. used the P_43_ promoter to overexpress the Cas9n protein in order to create a CRISPR/Cas9n gene editing system. The method can achieve 100% editing efficiency for a single gene, 11.6% editing efficiency for two genes, and 79.0% editing efficiency for a large fragment gene (42.7 KB) [29].

#### 2.2.2. P_Shuttle-09_

P_Shuttle-09_, developed in *B*. *subtilis*, is a potent promoter that is eight times more potent than the P_43_ promoter [35]. Additionally, P_Shuttle-09_ is frequently employed in *B*. *licheniformis* as a fundamental promoter for TF-TF binding site research [24].

## 3. Inducible Promoters

Inducible promoters can intensify expression in response to specific effectors such as light, osmotic pressure, pH, carbohydrates, amino acids, antibiotics, etc. Inducible promoters are preferable to constitutive promoters in two scenarios: (1) biosensor development and optimisation and (2) toxic protein induction expression, such as Cas12a.

### 3.1. Sugar-Inducible Promoters

#### 3.1.1. Xylose-Inducible Promoter

The xylose operon consists of a bidirectional promoter P*_xylAB_*, and three structural genes: *xylA*, *xylB*, and *xylR*. In the presence of *xylR* deletion, *xylAB* becomes constitutively expressed. The XylR protein of *B*. *licheniformis* forms a complex with xylose, thereby reducing the affinity of the XylR protein to its target xylO [36] (Figure 3A,B). According to Li et al., glucose reduced xylose operon transcription by over 168 times and did not significantly correlate with glucose substrate concentration [37]. Furthermore, at high temperatures (25–42 °C), the transcription of xylose operons steadily rises [37]. Therefore, the promoter is induced by xylose but inhibited by glucose. One commonly utilised promoter in *B. licheniformis* is the xylose-inducible promoter. Below is an overview of recent real-world applications of this promoter in *B. licheniformis*.

Li et al. created a P*_xyl_*-regulated Cas9 protein expression cassette to develop a xylose-induced CRISPR/Cas9 gene editing system. The plasmid conversion rate can be increased from 0.1 cfu/μg to 2.42 cfu/μg DNA by xylose-induced regulation of Cas9 expression, compared to constitutive Cas9 protein expression. Following transformation, the gene editing efficiency with *amyL* (encoding maltoamylase) as the target gene can reach 70.9% with the addition of 0.5% xylose. The editing efficiency can be further increased to 97% by lowering the temperature to 20 °C [21].

It was discovered that the xylose promoter from *B*. *subtilis* had the greatest induction impact when expressing the TreS enzyme using xylose promoters from three different sources (*B. subtilis*, *B. licheniformis*, and *Bacillus megaterium*). The optimal growth conditions for the engineered strain, which carries a xylose promoter-controlled *treS* expression cassette from *B. subtilis*, are to add 1% xylose, 0.4% soybean flour, and 4% maltodextrin after 10 h of culture. The induction period should be 12 h, and the maximum enzyme activity should be 24.7 U/mL [38].

When glucose, fructose, and sucrose are used as carbon sources, the enzyme activity of engineered strains is greatly inhibited because the CcpA protein binds to the cre site (TGAAAGCGATTAAT) located near the −10 region in the xylose promoter. By mutating the CG at the cre site to AT, the enzyme activity of the engineered strain increased by 12 times under glucose conditions. Using the xylose promoter mutant as an expression element to induce the production of maltose amylase, the highest detectable amylase activity at 37 °C was 715.4 U/mL [9].

The xylose-inducible promoter (P*_xyl_*) was substituted for the lichenysin biosynthetic operon promoter in the mutant *B. licheniformis* WX02-Pxyllch [39]. It was discovered that adding 50 mM xylose had the best induction effect and that the yield of lichenysin extract exceeded 40 mg/L when xylose was added at varied doses (0 mM–100 mM).

#### 3.1.2. Acetoin/2,3-Butanediol-Inducible Promoter

The promoter P*_aco_*, derived from the *B. licheniformis* acetoin operon (*acoABCL*, *acuABC*), was described by Thanh et al. [40]. Acetoin and 2,3-butanediol were found to stimulate two promoters, but glucose significantly inhibited them. Acetoin and 2,3-butanediol are examples of overflow metabolites that *B. licheniformis* produces when it uses glucose and other substrates. Consequently, the acetoin and 2,3-butanediol generated when glucose is reduced can reactivate the promoter. TGAAAACGCTTAAT has been identified as the cre site in P*_aco_* and is a critical location for the glucose-mediated CCR impact. In *Bacillus*, AcoR has been shown to be a key transcription factor regulating P*_aco_*. The deletion of the *acoR* gene prevents *Bacillus* from utilizing acetoin [41].

#### 3.1.3. Mannitol-Inducible Promoter

A mannitol-induced expression system was created by Xiao et al. using the *B. licheniformis* mannitol operon, which consists of the structural gene *mtlAFDR* and two promoters, P*_mtlA_* and P*_mtlR_* [42]. Mannitol can activate both P*_mtlA_* and P*_mtlR_*, with P*_mtlA_* exhibiting greater induction activity. P*_mtlA_* can also be induced by sorbitol, mannose, and arabinose, in addition to mannitol. Sorbitol is the best inducer when employing this promoter to express maltose amylase because it has the greatest induction effect. Two key TF-binding sites can be found in P*_mtlA_*: cre (TGTAAGCGTTTTTAA) and MtlR box (TTGTCA-cacggctcc-TGCCAA). The CcpA protein binds to the P*_mtlA_* cre site in the presence of glucose, blocking the promoter’s activity. It is possible to significantly reduce the CCR effect of glucose on P*_mtlA_* by changing the CG in the cre site to AT.

Recently, mannitol has received widespread attention as a marine carbon source [43]. Mannitol is the main carbon source for third-generation renewable biomass—seaweed biomass hydrolysate [44]. The development of mannitol induction systems is a way to apply third-generation renewable biomass to synthetic biology.

#### 3.1.4. Trehalose-Inducible Promoter

The three structural genes *treA*, *treP*, and *treR*, as well as the promoter P*_treA_*, comprise the *B. licheniformis* trehalose operon. The CcpA and TreR proteins control the activity of P*_treA_*. TreR box (TTGTATATACAA; ATGTATATACAA) and cre (TGAAAGCGCTATAA) are essential components in P*_treA_’*s response to transcription factors. Trehalose induces P*_treA_*, while glucose, fructose, sucrose, and mannose inhibit it. The CCR impact mediated by glucose is lessened when the *ccpA* gene is absent [45].

#### 3.1.5. Rhamnose-Inducible Promoter

Sugars can act as carbon suppliers and inducers for cells. As a result, using quick-acting carbon sources as inducers for induction expression systems—such as glucose and fructose—is challenging. *B. licheniformis* cells need 36 h to deplete 20 g/L of rhamnose, whereas they only need 9 h to do the same with glucose [46]. The promoter P*_rha_*, derived from the *B. licheniformis* rhamnose operon, was described by Xue et al. Rhamnose induces P*_rha_*, but glucose, mannitol, xylose, and sorbitol do not. The activity of the promoter is positively associated with increased concentration when rhamnose is added at a concentration of 0–20 g/L. This promoter increases the strain’s recombination efficiency by 105 times by controlling the expression of the *Bacillus* bacteriophage recombinant enzyme RecT. Future developments will see further expansion of rhamnose promoters in *B. licheniformis*.

#### 3.1.6. Mannose-Inducible Promoter

Zhang et al. used *B*. *licheniformis’*s endogenous mannose promoter P*_man_* to create a mannose-induced CRISPRi system. The system’s efficacy was confirmed by an 84% downregulation of transcription upon the addition of mannose, as measured by urease, the reporter protein. Downregulating the overflow metabolites 2,3-butanediol and acetic acid by 38% and 26%, respectively, was achieved by controlling the transcription level of the global transcription factor CodY, which encodes the gene *codY*, by 10%–75% [47].

#### 3.1.7. Lactose-Inducible Promoter

The lactose-inducible promoter P*_lac_* is derived from the lactose operon of *B. licheniformis* and is a bidirectional promoter [48]. This promoter is located between the *lacR* gene and *lacDCAB* in the genome. The lactose minimal inducing unit was identified as a 38 bp palindrome sequence Box1. CcpA, LacR, and TnrA proteins can all bind to Box1, thereby regulating the activity of P*_lac_*.

#### 3.1.8. IPTG-Inducible Promoter

An IPTG-inducible expression vector in *Bacillus* is the pHT01 plasmid. In order to produce isoprene, Gomma et al. employed this vector to overexpress the Kudzu isoprenoid synthase (*kIspS*) gene in *B. licheniformis* DSM 13. After 48 h at 37 °C and 0.1 mM IPTG incubation, this resulted in the production of 437.2 μg/L (249 μg/L/OD) isoprene [49].

### 3.2. Nitrogen-Inducible Promoters

#### Ammonia-Inducible Promoter

Shen et al. selected six genes (*copA*, *sacA*, *ald*, *pdbX*, *plP*, and *dfP*) that showed the largest transcriptional upregulation in the presence of ammonia based on transcriptome data under ammonia deficiency and ammonia presence. The activities of these six genes’ promoters were examined using *amyL* as the reporter gene. The findings demonstrated that five promoters had ammonia-induced activity, with the exception of the promoter P*_dfP_*. P*_plP_* had the highest ammonia induction value, but P*_sacA_* had the largest ammonia induction range (0–75%). Adding ammonia has two advantageous effects: (1) it balances the pH of the fermentation broth, and (2) it acts as an inducer to boost the promoter’s activity [50]. Sucrose metabolism is carried out by the SacA enzyme, which is encoded by the *sacA* gene of *B*. *licheniformis* [51]. The promoter’s ammonia-induced activity suggests that the SacA enzyme may be crucial for the conversion of carbon to nitrogen.

### 3.3. Antibiotic-Inducible Promoter

#### Tetracycline-Inducible Promoter

He et al. investigated the impact of the dal expression level on maltose amylase by overexpressing the dal gene using three different promoters (P_43_, P*_dal_*, and P_tet_). The amylase activity of the DAL expression system induced by tetracycline was the highest, reaching 155 U/mL, which was 27% higher than the control strain [52].

### 3.4. Auto-Inducible Phosphate-Controlled Promoter

Promoters originating from the phytase gene (*phyL*) of *B. licheniformis* were unearthed and identified by Trung et al. Promoter activity is markedly increased in the presence of a phosphate constraint, allowing foreign genes *amyE* and *xynA* to be expressed efficiently [53]. The two PhoP binding motifs found in the promoter, TTTACA and TTTTCA, suggest that the PhoPR two-component system regulates the promoter. Phytase catalyses the breakdown of phytate, releasing a range of lower isomers of myoinositol phosphates [54]. As a result, phytase sodium can also stimulate and control the system, and at a dose of ≤5 mM, the promoter can be significantly induced.

### 3.5. Environmental-Inducible Promoters

#### 3.5.1. Salt-Inducible Promoter

Under 1.3 M NaCl conditions, *B. licheniformis* DSM 13T entirely inhibits growth, and it can withstand 1 M NaCl. The findings of the combined transcriptome and Northern blot analyses indicate that the genes *proH*, *proJ*, and *proAA* co-transcribe as osmotic-inducible operons. The promoter P*_pro_* regulates this operon’s transcription. The reporter gene *treA* was fused with the promoter P*_pro_*, and it was discovered that 0.4 M NaCl could significantly stimulate gene expression. The promoter’s activity and NaCl have a positive correlation in the region of 0–1.0 M NaCl [55]. Furthermore, in *B. subtilis*, the *pro* operon is in charge of the metabolism of proline degradation and its reaction to osmotic pressure is connected to proline biosynthesis [56,57]. Proline is also one of the ways that plants survive when they are under salt stress [58]. To encourage its use in synthetic biology, the molecular mechanism regulating this promoter’s osmotic pressure can be examined in the future.

Guo et al. discovered that 6% NaCl activated genes linked to increasing glutamate production when comparing the transcriptome data of *B. licheniformis* WX-02 under normal and high-salt conditions (NaCl 6%) [59]. Additionally, Binda et al. demonstrated a linear relationship between *B*. *licheniformis’*s gamma glutamyl transpeptidase production and the concentration of NaCl [60]. These genes’ expression is regulated by promoters, which may also be salt-inducible promoters.

#### 3.5.2. pH-Inducible Promoter

As of right now, *B. licheniformis* pH-induced promoters have not been clearly reported. ParK et al. discovered that the expression levels of genes associated with the metabolism of fatty acids, malic acid, and branched chain amino acids are pH-related [61] based on transcriptomics and metabolomics. It has been shown by Hornbaek et al. that B. licheniformis enhances acetoin production to neutralize pH in low-pH environments [62]. According to Wang et al., *B*. *licheniformis* poly-γ-glutamic acid (γ-PGA) had a maximum production of 36.26 g L^−1^ under alkaline stress, a 79% increase over the control group. The γ-PGA synthase genes *pgsB* and *pgsC*, together with their closely associated regulatory components *swrA* and *degU*, showed increases in transcription levels of 18.9, 31.2, 3.0, and 6.3 times, respectively [63]. Furthermore, pH controls how poisonous *B*. *licheniformis* DAS-2 is to arsenic [64]. The genes whose promoters are discussed in these papers may join the group of pH-responsive promoters.

#### 3.5.3. Temperature-Inducible Promoter

As of right now, *B. licheniformis* temperature-induced promoters have not been clearly reported. The metabolomics and proteomics of *B*. *licheniformis* at high temperatures will be the basis for many studies on heat-induced promoters in the future [65,66,67]. Lo et al. established that *B*. *licheniformis’*s HtPG protein is a heat shock protein and that a high temperature may activate its promoter [68].

## 4. Quorum Sensing Promoter

The quorum sensing promoter P_lan_ is found in the gene cluster for lanthanide biosynthesis, and the Agr QS system in *Staphylococcus aureus* and its upstream gene cluster LanCBDA share some similarities. As a result, it is believed that P_lan_ is part of an Agr-like QS system. Between 0 and 24 h, the promoter’s activity is incredibly low, but between 36 and 48 h, it increases dramatically. After 48 h, the P_lan_ promoter’s activity is 75 times higher than that of the P_43_ promoter when the green fluorescent protein is used as the reporter gene [69].

## 5. Promoter Engineering in *B. licheniformis*

The accurate control of gene expression related to biosynthetic pathways through highly intense and modifiable promoters is a prerequisite for creating strains of any host. Few promoters completely fulfil the requirements for gene circuits, making modified promoters essential for synthetic biology. Several promoter techniques have been developed and proposed in *B*. *licheniformis* to enhance the adjustability and output threshold. These tactics include (1) hybrid promoter engineering; (2) transcription factor-based induced promoter engineering; and (3) RBS engineering (Figure 3C,D).

### 5.1. Hybrid Promoter Engineering

The artificial assembly of several possible promoters, or hybrid promoter engineering, primarily takes two forms: (1) combining promoters A and B; (2) combining promoter A with part of promoter A. A+B is often implemented by manually assembling the two promoters. Three artificial hybrid promoters, P*_ylB_*-P_43_, P*_gsiB_*-P_43_, and P*_ykzA_*-P_43_, were created by hybridising the promoters (P*_ylB_*, P*_gsiB_*, and P*_ykzA_*) with the P_43_ promoter. When compared to the P_43_ promoter, the artificial promoter P*_ykzA_*-P_43_ performs optimally, increasing the expression of the green fluorescent protein, the red fluorescent protein, and amylase by 1.72, 3.46, and 1.85 times, respectively [70].

UP elements interact with the alpha subunit of RNA polymerase (RNAP) and are widely distributed in non-coding areas. The action of natural promoters can be increased by mixing them with UP components. The natural strong promoter plan’s UP components have been described. Li et al. created an artificial UP element called UP5-2P based on this UP element. The artificial UP components and promoter can be combined to improve the promoter activity by over eight times [69].

Prokaryotic mRNAs possess a 5′-untranslated region (5′-UTR) that includes the Shine–Dalgarno (SD) sequence and an optional translation enhancer sequence. This region is crucial for both translation initiation and RNA stability [71]. Xiao et al. developed a portable 5′-UTR sequence that forms a hairpin structure immediately upstream of the open reading frame, comprising approximately 30 nucleotides. This 5′-UTR can enhance the production of eGFP by roughly 50-fold by optimising the free energy of folding and shows strong adaptability to other target proteins, including RFP, nattokinase, and keratinase [72].

### 5.2. Transcription Factor-Inducible Promoter Engineering

Native promoters and regulatory elements are frequently selected for regulating gene expression in *B. licheniformis*. These regulatory elements, which determine whether transcription is activated or repressed, are often controlled by transcription factors. Artificial promoters with higher performance than native promoters can be created by engineering transcription factor binding sites into constitutive promoters. The primary transcription factor controlling the mannitol operon is the MtlR protein, which modifies its affinity for the mannitol promoter based on its phosphorylation state [73]. Using the constitutive promoter P_shutle09_, Xiao et al. incorporated the MtlR box (TTGTCA-cacggctcc-TGCCAA) to create an artificial promoter inducible by mannitol [74]. This suggests that effectors can add transcription factor binding sites to constitutive promoters to influence the transcription process. In several species, the deletion of transcription factor binding sites has been observed to convert naturally occurring inducible promoters into constitutive promoters [75].

The nitrogen global transcription factor GlnR was recently used to create a sorbitol-activated nitrogen metabolism regulation system. The GlnR binding site motif in *B. licheniformis* has been identified as “TGTNAN7TNACA” [24]. Sorbitol can achieve up to 99% suppression of the target protein by regulating the binding of GlnR and the GlnR box through the use of the promoter P*_mtlA_*. This technique can be used to reroute carbon overflow metabolism, leading to a 2.6-fold increase in acetic acid synthesis and a 79.5% decrease in acetoin production.

A malic acid-induced biosensor was created through the production of malic acid reactive TF (MalR) from *B. licheniformis*. Zhang et al. reported that all six promoter groups involved in malic acid metabolism genes (P*_cimH_*, P*_maeA_*, P*_maeN_*, P*_mdH_*, P*_malA_*, and P*_ytsJ_*) were sensitive to malic acid. P*cimH* had the highest malate response value [76]. Malic acid and the transcription factor MalR control the activity of P*_cimH_*. The minimum inducible functional unit of malic acid was determined to be “TTAATTAGTTAAATAACTCAGAGCAAAGGGATAACAAAAA” (MalR box) through in vitro and in vivo fluorescence tests. The activity of the hybrid promoter created by assembling the MalR box into the constitutive promoter P_shutle09_ responds linearly to malic acid concentrations ranging from 5 to 15 g/L. The biosensor based on this promoter can be used to screen *B. licheniformis* for malic acid synthesis by providing standardised components for biosynthesis.

### 5.3. RBS Engineering

The ribosome binding site (RBS) directly impacts protein abundance and quality by influencing translation fidelity and efficiency [77]. Rao et al. constructed an RBS library in *B. licheniformis*, providing incremental regulation of expression levels over a 104-fold range [78]. Zhang et al. engineered a novel mRNA leader sequence containing multiple RBSs, which could initiate translation from multiple sites, vastly enhancing translation efficiency in *B. licheniformis* [79].

These artificial promoters can better detect the dynamic changes in intracellular metabolite concentrations and balance the competition between product synthesis and cellular metabolism.

## 6. Concluding Remarks and Outlook

With recent advances in synthetic biology, *B*. *licheniformis* has become increasingly popular. Although there have been some satisfactory results in *B. licheniformis*, such as (1) the recombinant *B. licheniformis* designed by Zhou et al. producing 11.33 g/L nicotinamide riboside (NR) [80] and (2) the engineering strain of B. licheniformis designed by Zhan et al. producing 5.16 g/L 2-phenylethanol using molasses as a carbon source [81], the promoter remains a major factor limiting the application of *B. licheniformis*. Promoters are genetic elements that refine gene expression, and promoter engineering maximises the production of target compounds by regulating overall metabolic balance. Despite the significant progress made through promoter engineering strategies, challenges still limit their application in *B. licheniformis*. The main reasons include (1) the incomplete characterisation of endogenous promoters; (2) the few endogenous promoters developed; (3) the lack of understanding of dynamic control components.

One of the Gram-positive bacteria that has been investigated the most is *B*. *subtilis*. Currently, *B*. *subtilis* has a more effective gene editing system than *B. licheniformis*. Wu et al.‘s CRISPR/Cpf1, for instance, has a 100% effectiveness rate in achieving single-gene insertion, six-site mutations, and dual-gene knockout [82]. By creating a new generation base editor with an extended editing window, Hao et al. were able to significantly increase *B*. *subtilis* cell evolution screening efficiency [83]. Furthermore, Guo et al. prevented potential antibiotic contamination by creating plasmid-free, stable *B*. *subtilis* [84]. The various forms of promoters found in *B. subtilis* serve as the foundation for the above investigations. There are not as many studies on *B. licheniformis*’ promoters as there are on *B. subtilis*. Further development of promoters is a necessary step toward broadening the industrial applicability of *B. licheniformis*.

The quorum sensing promoter can balance bacterial production and product synthesis well and is therefore favoured by metabolic engineering. Currently, there is a lack of reported quorum sensing promoters in *B. licheniformis*, with only P_lan_ available. Further development should be carried out on the quorum sensing promoter of *B. licheniformis*.

The application scenarios of sugar alcohol-inducible promoters are extensive, encompassing CRISPR gene editing systems and high-value enzyme production. In industrial contexts, the cost of inducers is a critical factor in evaluating the efficacy of inducible promoters. Maltose, being inexpensive, has not yet been exploited in *B*. *licheniformis*. In *B. subtilis*, the maltose-inducible promoter has been developed into a robust induction expression system [85]. Therefore, by analysing the maltose metabolism pathway and exploring potential maltose promoters, a maltose-induced expression system suitable for *B. licheniformis* can also be developed. A significant obstacle in the application of sugar alcohol-inducible promoters is the glucose-mediated carbon catabolite repression (CCR) effect. Understanding the molecular mechanisms underlying CCR is crucial for enhancing the performance of sugar alcohol-induced promoters.

Promoters induced by amino acids can be utilised to develop specific amino acid biosensors for the biosynthesis of amino acids [86]. Currently, there are few reports on amino acid-inducible promoters in *B. licheniformis*. One approach to developing amino acid-inducible promoters is to extract promoters from amino acid operons. For example, the proline-inducible promoter in *B. subtilis* is derived from the proline operon [87]. By analysing the amino acid metabolism pathways and regulatory mechanisms in *B. licheniformis*, suitable amino acid-inducible promoters can be developed.

High-temperature fermentation can significantly reduce contamination and condensation costs [88]. Industrially, fermentation processes using thermophiles (above 45 °C) are defined as high-temperature fermentation [89]. *B. licheniformis* is an excellent high-temperature platform strain capable of rapid growth at 50 °C [18,19]. Overexpression of the heat-resistant gene *groES* (originating from *B. licheniformis*) in *B. subtilis* can enhance its heat tolerance [90]. However, the lack of available promoters for *B. licheniformis* at high temperatures currently limits its application. In the future, high-temperature transcriptome data can be used to develop high-temperature-responsive promoters for *B. licheniformis*.

Anaerobic fermentation offers unique advantages such as low energy consumption and reduced pollution risk, especially in biofuel production [91]. *B. licheniformis* is better adapted to anaerobic growth than *B. subtilis* [92]. The Fnr protein is the main transcription factor in *B. licheniformis* under anaerobic conditions [93]. Currently, there are no reports of anaerobic promoters in *B. licheniformis*. By analysing the regulatory mechanism of Fnr, artificial anaerobic promoters can be developed for metabolic engineering under anaerobic conditions.

The pace of biological engineering and discovery is being substantially accelerated by the merging of artificial intelligence (AI) with synthetic biology. Synthetic biology and newly created AI tools have the potential to work wonders for intelligent manufacturing in the fourth industrial revolution (industry 4.0) in the years to come [94]. In the future, *B. licheniformis* promoters may be engineered using machine learning based on deep learning techniques. Accelerating the engineering of *B. licheniformis* promoters can be achieved by de novo TF design and by calculating and predicting the ideal TF recognition site.

Promoter engineering relies on a clear understanding of the interaction between transcription factors and promoters. It is crucial to further expand the promoter library of *B. licheniformis* and develop promoters suitable for various biosynthesis scenarios hosted by *B. licheniformis*.

## Figures and Tables

**Figure 1 microorganisms-12-01693-f001:**
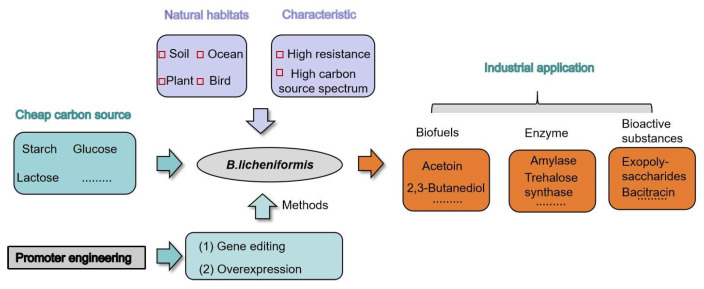
The application of *B. licheniformis.*

**Figure 2 microorganisms-12-01693-f002:**
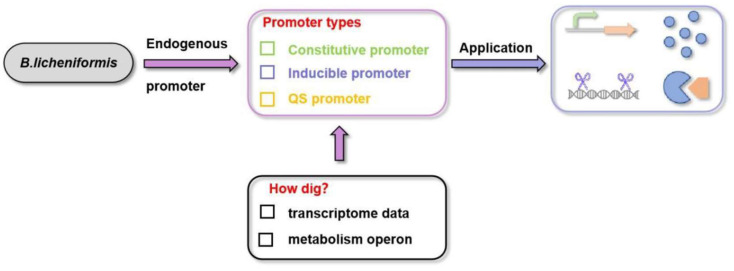
Promoter in *B.licheniformis.*

**Figure 3 microorganisms-12-01693-f003:**
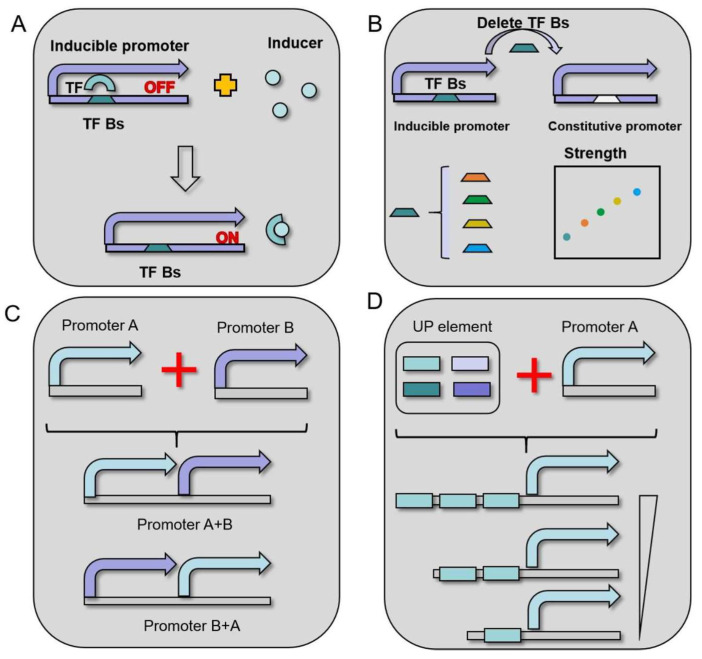
Promoter strategies in *B. licheniformis.* (**A**) Natural inducible promoter (**B**) Transcription factor-inducible promoter engineering (**C**) Hybird promoter engineering (**D**) UP element engineering.

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
