# Peer review of "Advancing Bacillus licheniformis as a Superior Expression Platform through Promoter Engineering"

_microorganisms, 2024, doi:10.3390/microorganisms12081693_

Round 1
Reviewer 1 Report
Comments and Suggestions for Authors
The review is well-focused and devoted to promoters applicable to the species B. licheniformis. The authors highlight the broad application of the species and the need for genetic engineering. The work involves a lot of data, but it is not well organized. I have the following suggestions for improving performance:
1. Figure 1 is very large and not informative enough. Include the industrial applications of B. licheniformis, which are numerous, from biofuel production to enzymes for the food industry, exopolysaccharides and so on.
2. A table with the applicable B. licheniformis vectors (shuttle, integrative and others) must be made. Promoters are always part of a vector and of a particular cloning system, the promoter cannot be considered separately.
3. The numbering of the chapters in the text must be improved. Such texts as "Introduction", "Constitutive Promoter" (which should be "Constitutive Promoters"), and the rest of the text in bold should be given with numbering. There is no need of different chapters to describe the promoters, which are induced by different sugars - you should merge them into one: "The "sugar-inducible promoters" chapter.
4. Among the inducible promoters, I did not see those that are induced by IPTG and also different types of environmental stress (temperature, pH, salts)
5. Please, add a table that reveals the practical application of the promoters of different types. For example, in the cloning of which enzymes or other genes these promoters were used and, what efficiency they achieved.
As an example, you can use two good recent articles:
https://doi.org/10.1016/j.synbio.2023.03.008
https://doi.org/10.3390/fermentation10010050
Comments on the Quality of English LanguageMinor editing of English language required.
Author Response
Reviewer #1: The review is well-focused and devoted to promoters applicable to the species B. licheniformis. The authors highlight the broad application of the species and the need for genetic engineering. The work involves a lot of data, but it is not well organized. I have the following suggestions for improving performance:
- Figure 1 is very large and not informative enough. Include the industrial applications of B. licheniformis, which are numerous, from biofuel production to enzymes for the food industry, exopolysaccharides and so on. Answer: Thanks for your correction and suggestion. The Figure 1 has been revised in the manuscription.
- A table with the applicable B. licheniformis vectors (shuttle, integrative and others) must be made. Promoters are always part of a vector and of a particular cloning system, the promoter cannot be considered separately.
Answer: Thanks for your correction and suggestion. The table of the vectors have been added in the manuscription.
Table 1 The vectors in B.licheniformis
Vectors
Describe
Ref
Gene expression
pHY-PLK300
E. coli/Bacillus shuttle vector, ApR /TetR
42
pWB980
Bacillus vector KanR
42
pHT43
E. coli/Bacillus shuttle vector,ChlR
47
Gene editing
pNZT1
Temperature-sensitive plasmid, E. coli/Bacillus shuttle vector,TetR
34
pKVM
E. coli/Bacillus shuttle vector,TetR
95
- The numbering of the chapters in the text must be improved. Such texts as "Introduction", "Constitutive Promoter" (which should be "Constitutive Promoters"), and the rest of the text in bold should be given with numbering. There is no need of different chapters to describe the promoters, which are induced by different sugars - you should merge them into one: "The "sugar-inducible promoters" chapter.
Answer: Thanks for your correction and suggestion. The format has been corrected in the latest manuscript.
- Among the inducible promoters, I did not see those that are induced by IPTG and also different types of environmental stress (temperature, pH, salts)
Answer: Thanks for your correction and suggestion. The IPTG-inducible promoter and environmental stress-inducible promoters have been added in the manuscription.
3.1.8 IPTG-Inducible Promoter
An IPTG inducible expression vector in Bacillus is the pHT01 plasmid. In order to produce isoprene, Gomma et al. employed this vector to overexpress the Kudzu isoprenoid synthase (kIspS) gene in B. licheniformis DSM 13. 48 hours at 37 °C and 0.1 mM IPTG incubation resulted in the production of 437.2 μg/L (249 μg/L/OD) isoprene [49].
3.5 Environmental-Inducible Pormoter
3.5.1 Salts-Inducible Promoter
Under 1.3 M NaCl conditions, B. licheniformis DSM 13T entirely inhibits growth, and it can withstand 1 M NaCl. The findings of the combined transcriptome and Northern blot analyses indicate that the genes proH, proJ, and proAA cotranscribe as osmotic inducible operons. The promoter Ppro regulates this operon's transcription. The reporter gene treA was fused with the promoter Ppro, and it was discovered that 0.4M NaCl could significantly stimulate gene expression. The promoter's activity and NaCl have a positive correlation in the region of 0-1.0M NaCl [55]. Furthermore, in B. subtilis, the pro operon is in charge of the metabolism of proline degradation and its reaction to osmotic pressure is connected to proline biosynthesis [56–57]. Proline is also one of the ways that plants survive when they are under salt stress [58]. To encourage its use in synthetic biology, the molecular mechanism regulating this promoter's osmotic pressure can be examined in the future.
Guo et al. discovered that 6% NaCl activated genes linked to increasing glutamate production when comparing the transcriptome data of B. licheniformis WX-02 under normal and high salt conditions (NaCl 6%) [59]. Additionally, Binda et al. demonstrated a linear relationship between B. licheniformis's gamma glutamyl transpeptidase production and the concentration of NaCl [60]. These genes' expression is regulated by promoters, which may also be salt-inducible promoters.
3.5.2 pH-Inducibe Promoter
As of right now, B. licheniformis pH-induced promoters have not been clearly reported. ParK et al. discovered that the expression levels of genes associated to the metabolism of fatty acids, malic acid, and branched chain amino acids are pH-related [61] based on transcriptomics and metabolomics. It has been shown by Hornbaek et al. that B. licheniformis enhances acetoin production to neutralize pH in low pH environments [62]. According to Wang et al., B. licheniformiss poly-γ-glutamic acid (γ-PGA) had a maximum production of 36.26 g L-1 under alkaline stress, a 79% increase over the control group. The γ-PGA synthase genes pgsB and pgsC, together with their closely associated regulatory components swrA and degU, showed increases in transcription levels of 18.9, 31.2, 3.0, and 6.3 times, respectively [63]. Furthermore, pH controls how poisonous B. licheniformis DAS-2 is to arsenic [64]. The genes whose promoters are discussed in these papers may join the group of pH responsive promoters.
3.5.3 Temperature-Inducible Promoter
As of right now, B. licheniformis temperature-induced promoters have not been clearly reported. The metabolomics and proteomics of B. licheniformis at high temperatures will be the basis for many studies on heat-induced promoters in the future [65-67]. Lo et al. established that B. licheniformis's HtPG protein is a heat shock protein and that a high temperature may activate its promoter [68].
- Please, add a table that reveals the practical application of the promoters of different types. For example, in the cloning of which enzymes or other genes these promoters were used and, what efficiency they achieved.
As an example, you can use two good recent articles:
https://doi.org/10.1016/j.synbio.2023.03.008
https://doi.org/10.3390/fermentation10010050
Answer: Thanks for your correction and suggestion. The table 3 has been added in the manuscription.
Table 3 The application of promoters in B.licheniformis
Promoters
Application scenarious
beneficial effects
Ref
PbacA
Pulcherriminic acid production
507.4 mg/L pulcherriminic acid
29
Increase glycerol consumption
an 18.8% rise in glycerol consumption over the wild strain
30
Short-chain fatty acids production
8.37 g/L of SBCFAs in a 5L fermentation tank
31
P43
Nattokinase production
35.60 FU ml-1 of nattokinase activity
33
The ccpA gene overexpression
The co-uptake between glucose and xylose
34
PxylA
Xylose-inducible CRISPR/Cas9 system
The editing efficiency can reach 97%
21
The expression of trahalose synthase
The maximum enzyme activity was 24.7 U/mL
38
The expression of amylase
The highest detectable amylase activity at 37°C was 715.4 U/mL
9
The lichenysin biosynthetic
The production of lichenysin exceeded 40 mg/L
39
Prha
The expression of the Bacillus bacteriophage recombinant enzyme RecT
The recombination efficiency increased by 105 times
46
Pman
Mannose-inducible CRISPRi system
The transcription inhibition efficiency can reach 84%
47
Reviewer 2 Report
Comments and Suggestions for Authors
Dear authors, the manuscript "Advancing Bacillus licheniformis as a Superior Expression Platform through Promoter Engineering" is quite interesing and worth investigation. Please see some comments below:
1- Please double-chekc formatting and grammar, for instance . Natural habitats for B. li-cheniformis include instead of Natural habitats for B. li-cheniformis include
2- The discussion is quite good (deep). Have you considered any details with in silico information? I mean the strategy, primers, etc?
3- It is a simple molecule. Is it possible to do it synthetic?
4- Regarding the B. subtilis? There is more information abour it. What are the advantages, when compared B. licheniformis?
Regards
Author Response
Reviewer #2: Dear authors, the manuscript "Advancing Bacillus licheniformis as a Superior Expression Platform through Promoter Engineering" is quite interesing and worth investigation. Please see some comments below:
- Please double-chekc formatting and grammar, for instance . Natural habitats for B. li-cheniformis include instead of Natural habitats for B. li-cheniformis include
Answer: Thanks for your correction and suggestion. We have corrected it in the manuscription.
- The discussion is quite good (deep). Have you considered any details with in silico information? I mean the strategy, primers, etc?
Answer: Thanks for your correction and suggestion. We added the discussion of silico information in the manuscription.
The pace of biological engineering and discovery is being substantially accelerated by the merging of artificial intelligence (AI) with synthetic biology. Synthetic biology and newly created AI tools have the potential to work wonders for intelligent manufacturing in the fourth industrial revolution (industry 4.0) in the years to come [94]. In the future, B. licheniformis promoters may be engineered using machine learning based on deep learning techniques. Accelerating the engineering of B. licheniformis promoters can be achieved by de novo TF design and by calculating and predicting the ideal TF recognition site.
- It is a simple molecule. Is it possible to do it synthetic?
Answer: The promoter length is usually below 200 bp and can be obtained through synthesis.
- Regarding the B. subtilis? There is more information abour it. What are the advantages, when compared B. licheniformis?
Answer: Thanks for your correction and suggestion.
One of the Gram-positive bacteria that has been investigated the most is B. subtilis. Currently, B. subtilis has a more effective gene editing system than B. licheniformis. Wu et al.'s CRISPR/Cpf1, for instance, has a 100% effectiveness rate in achieving single gene insertion, six site mutations, and dual gene knockout [82]. By creating a new generation base editor with an extended editing window, Hao et al. were able to significantly increase B. subtilis cell evolution screening efficiency [83]. Furthermore, Guo et al. prevented potential antibiotic contamination by creating plasmid-free, stable B. subtilis [84]. The various forms of promoters found in B. subtilis serve as the foundation for the above investigations. There are not as many research on B. licheniformis' promoter as there are on B. subtilis. Further development of promoters is a necessary step towards broadening the industrial applicability of B. licheniformis.
Round 2
Reviewer 1 Report
Comments and Suggestions for Authors
I am fully satisfied by the corrections.
Comments on the Quality of English LanguageThe English is fine.